# CAReFuseNet: Cross Attention Fusion Network for Referring Camouflaged Object Detection

## Abstract

The Referring Camouflaged Object Detection (Ref-COD) task aims to generate a binary segmentation mask to detect camouflaged objects of a specified category in an image, guided by reference image(s) containing salient example(s) of the target object of the same category. With only a few methods (e.g., R2CNet and UAT) proposed to date, Ref-COD remains challenging due to the similarity of camouflaged objects to their backgrounds and substantial feature gaps with salient references. At the same time, recent state-of-the-art approaches often rely on heavy transformer-based encoder–decoder stacks or large frozen vision backbones, resulting in substantial parameter footprints that hinder efficient deployment. This work proposes 'CAReFuseNet'[1], a novel framework featuring a cross-attention based reference feature fusion module that effectively extracts reference-conditioned feature representations from camouflaged images while targeting parameter efficiency. The proposed CAReFuse module leverages global interactions between reference and camouflaged image features via cross-attention, but constrains all fusion and decoding operations to a lower dimensional feature space and employs a lightweight convolutional decoder. Combined with a frozen Ref-Image Encoder, this design yields a compact Ref-COD model without sacrificing accuracy. Extensive experiments on the R2C7K dataset show that our method surpasses state-of-the-art, while using significantly fewer parameters. Further evaluations across multiple backbone architectures, including Swin Transformer, ConvNeXt, EfficientNet, and ResNet, demonstrate that the proposed reference feature fusion module provides a general and parameter-efficient building block for the referring camouflaged object detection task.

## 1 Introduction

Camouflaged Object Detection (COD) (Fan et al., 2020a), which aims to identify the objects that are seamlessly blended into their background, is a challenging yet useful computer vision task. The intrinsic similarities between the target object and the background, such as indistinguishable texture and ambiguous object boundaries, make COD a difficult task for computer vision systems. Owing to its real-world applications ranging from medical image segmentation (e.g., polyp segmentation (Fan et al., 2020b), lung infection segmentation (Fan et al., 2020c; Wu et al., 2021)) to video surveillance (e.g., camouflaged person or object detection), the COD task is gaining increased attention in the computer vision community (Lei et al., 2025; Le et al., 2025; Ren et al., 2025; Chen et al., 2025). One novel task setting of the COD problem is the Referring Camouflaged Object Detection (Ref-COD) (Zhang et al., 2025), where a reference image (or a set of reference images) containing a salient example of the target object is used to guide the detection of the camouflaged object in another image. Similar to Referring Image Segmentation (Hu et al., 2016; Lee et al., 2025), this task setting proves invaluable, especially in scenarios where one knows in advance what target object category one is looking for in the camouflaged images, as well as when the object is uncommon. However, there can be variation in pose, appearance, shape and size between the salient object instance in the reference images and the camouflaged objects in the input camouflaged image (see Figure 1). Zhang et al. (2025), who introduced the Ref-COD task, have proposed a dual-branch framework, dubbed R2CNet,

---

[1]Code is available in supplementary material.

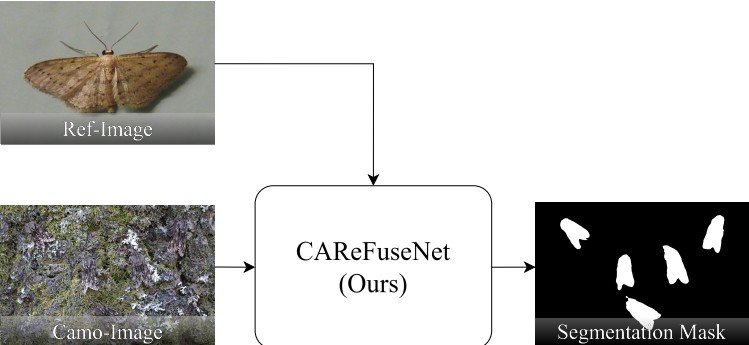

Figure 1: Despite variation in pose, appearance, shape and size between reference and target object instances, our CAReFuseNet segments camouflaged objects in the input image under the guidance of the salient reference image.

to perform the Ref-COD task by fusing the features extracted from a set of reference images with the feature representations of the input camouflaged image.

Following R2CNet, only a few other methods(e.g., UAT (Wu et al., 2025) and CIRCOD (Gupta et al., 2025)) emerged for the Ref-COD task. While UAT advanced the state of the art, it relies on heavy transformer-based encoder–decoder stacks. For instance, UAT employs a relatively deep transformer encoder–decoder architecture for reference feature integration and uncertainty modeling. Such designs yield high performance but come with substantial trainable and total parameter footprints, which can hinder deployment in resource-constrained environments or latency-sensitive applications. This exposes an important gap: there is a need for Ref-COD frameworks that retain competitive performance while being substantially more parameter-efficient.

Towards addressing this gap, this paper presents a novel framework, dubbed 'Cross Attention Fusion Network for Referring Camouflaged Object Detection' (CAReFuseNet). The framework is designed to extract reference-conditioned feature representations of the input camouflaged image, which are subsequently used to generate a binary foreground map to segment the target camouflaged object in the image. It independently extracts feature representations of the reference image(s) and of the camouflaged image, and then fuses them to align the camouflaged image representations with the reference target object. To perform this reference feature fusion, a cross-attention based reference feature fusion module, dubbed 'CAReFuse', is proposed. Specifically, two submodules are designed, namely the Cross-Attention Fusion (CAF) and the Multi-Scale Fusion (MSF) modules. The CAF module fuses the reference features with the multi-scale features extracted from the camouflaged image by performing cross-attention between them, thereby enabling global interactions between reference and camouflaged features. The MSF module then combines the reference-conditioned multi-scale features (the output of the CAF module) into final feature representations. Crucially, all fusion operations are carried out in a unified low-dimensional feature space, and a lightweight convolutional decoder is used for segmentation, leading to a compact overall architecture.

Extensive empirical evaluation is conducted to demonstrate both the effectiveness and the parameter efficiency of the proposed framework. With only 25% as many trainable parameters and 52% as many total inference parameters as SOTA Uncertainty-Aware Transformer (UAT) (27.22M vs 111.60M trainable; 89.30M vs 173.64M total; see Table 1), the PVT-v2 instantiation of CAReFuseNet (Ours-P) surpasses UAT in terms of standard Ref-COD metrics. These results indicate that a carefully designed, low-dimensional cross-attention fusion module, coupled with a lightweight decoder (Sections 3.3.1 and 3.2), can yield superior performance to heavier transformer-based architectures at a substantially reduced parameter budget.

**Contributions.** To summarize, the contributions of this work are as follows:

- A novel framework for the Referring Camouflaged Object Detection task, dubbed CAReFuseNet, which demonstrates that a single-stage reference feature fusion in a low-dimensional feature space,

==highlight==paired with a light-weight segmentation decoder can yield superior performance to existing methods at a fraction of the parameter budget.==highlight==

- A new feature fusion module, composed of cross-attention reference fusion and multi-scale fusion and dubbed 'CAReFuse', is introduced to extract reference-conditioned feature representations of camouflaged images in a unified low-dimensional feature space.

- Extensive experiments showing that CAReFuseNet (Ours-P) achieves superior performance to state-of-the-art UAT, while using only one quarter as many trainable parameters and roughly half as many total inference parameters (see Table 1), establishing CAReFuseNet as a strong parameter-efficient baseline for Ref-COD and demonstrating its generality across multiple backbone architectures.

## 2 Related Work

We first review the advances in Camouflaged Object Detection (COD), in Section 2.1, highlighting the notable methods that have been proposed. Next, we discuss the emerging novel task setting called Referring Camouflaged Object Detection (Ref-COD) in Section 2.2, which is also the task studied in our work. Finally, we explore the cross-attention based feature fusion methods, in Section 2.3, as it pertains to reference-guided vision tasks.

### 2.1 Camouflaged Object Detection

One of the first works that attempted to solve the challenging problem of identifying camouflaged objects in confusing and deceptive scenes was proposed by Fan et al. (2020a), which paved the way for increasing interest in COD. Their work published a large-scale dataset, named COD10K, which contains images of camouflaged objects and their annotations, and a framework, called SINet (Fan et al., 2020a), to predict the binary segmentation mask for the input camouflaged image. Subsequently, several deep learning strategies emerged for COD, including multi-scale-context based strategies such as CamoFormer (Yin et al., 2024) and FSPNet (Huang et al., 2023), mechanism simulation based strategies such as ZoomNet (Pang et al., 2022) and PreyNet (Zhang et al., 2022), and multi-source information fusion strategies such as FEMNet (Zhong et al., 2022) and FEDER (He et al., 2023). A survey paper by Xiao et al. (2024) extensively presents all the works related to COD. Despite various strategies and methods proposed, COD still remains a formidable problem. Owing to the increased attention towards COD due to several real-world applications (Fan et al., 2020b;c; Tabernik et al., 2020; Le et al., 2020), some novel task settings of COD have also emerged. One novel setting of COD is Referring Camouflaged Object Detection.

### 2.2 Referring Camouflaged Object Detection

The Referring Camouflaged Object Detection (Ref-COD) task, proposed by Zhang et al. (2025), requires detecting the camouflaged objects of only a particular user-specified category rather than detecting all the camouflaged objects in the input image. Thus, Ref-COD requires reference images containing salient target objects, along with the input camouflaged image, to guide the detection of the target camouflaged object. This makes Ref-COD a more constrained and more practically useful task setting of COD. Zhang et al. (2025) assembled a large-scale dataset, called R2C7K (Zhang et al., 2025), containing 7K images (including camouflaged and reference images) covering 64 object categories in real-world scenarios. R2C7K remains the only large-scale dataset containing salient reference images along with camouflaged images, unlike several COD datasets that contain camouflaged images alone. Zhang et al. (2025) also proposed a framework, dubbed R2CNet (Zhang et al., 2025), which fuses reference image features with camouflaged image features in two stages using a Referring Mask Generation module followed by a Referring Feature Enrichment module. Our proposed framework achieves the reference feature fusion in a single stage through our novel feature fusion module, called the CAReFuse module.

Recent methods further improved state-of-the-art (SOTA) in referring camouflaged object detection. Wu et al. (2025) introduced Uncertainty-Aware Transformer (UAT) for Referring Camouflaged Object Detection, aggregating reference features with cross-attention and modelling token uncertainties via a probabilistic

decoder. While effective, UAT relies on a relatively heavy transformer encoder–decoder stack for visual reference integration and uncertainty modeling, whereas our method seeks to achieve better performance with a substantially more compact fusion and decoding design. Another method named CIRCOD (Gupta et al., 2025) was proposed, which adopted a co-saliency-inspired approach to perform the Ref-COD task. However, CIRCOD uses slightly different settings than R2CNet and UAT: a higher image size (*i.e.* $512 \times 512$) and a single reference image (*i.e.* $K = 1$), versus $352 \times 352$ and $K = 5$ for the latter methods.

### 2.3 Cross-Attention based Feature Fusion

Since the success of transformer architecture (composed of attention operations) for vision tasks (Dosovitskiy et al., 2021), cross attention is also being used for feature fusion in vision tasks. CrossViT (Chen et al., 2021) was one of the early works that used cross-attention to fuse image feature representations extracted at two different granularities. Shen et al. (2024) have used iterative cross-attention guided feature fusion for multi-spectral object detection. Li & Wu (2024) and Sun et al. (2025) have used cross-attention mechanism for cross-modal feature fusion between infrared and visible images. Cross-attention fusion has also been used for cross-domain feature fusion by Tripathi et al. (2020; 2024), where feature representations of hand-drawn sketches are fused with image features for the sketch-guided object localization task that detects and localizes the target objects as specified by a sketch query. While these applications demonstrate cross-attention's versatility across modalities and scales, its effectiveness in fusing salient target object features with camouflaged image features remains underexplored. Our work addresses this by proposing a cross-attention-based feature fusion method for Ref-COD, which demonstrably surpasses the existing Ref-COD approaches.

## 3 Methodology

We first describe the formulation of the problem in Section 3.1. Next, we explain the overall architecture of our proposed framework in Section 3.2. Subsequently, we present our novel CAReFuse module in Section 3.3, comprising of the Cross-Attention Fusion module in Section 3.3.1 and the Multi-Scale Fusion module in Section 3.3.2.

### 3.1 Problem Formulation

Ref-COD is a reference-guided foreground map prediction task and is formally defined as follows. For a given image containing camouflaged objects, termed $I^{camo} \in \mathbb{R}^{3 \times H \times W}$, and a given set of referring images containing a salient example of target object category $c$, denoted $\{I_k^{ref}\}_{k=1}^K, I_k^{ref} \in \mathbb{R}^{3 \times H \times W}$, the output of Ref-COD is a binary segmentation mask $M^{seg} \in \{0, 1\}^{1 \times H \times W}$ for the camouflaged objects of category $c$ in $I^{camo}$. Here, $H$ and $W$ represent height and width of the image, respectively. $M_{ij}^{seg} = 0$ indicates the pixel at position $(i, j)$ belongs to background, while $M_{ij}^{seg} = 1$ represents the pixel $(i, j)$ is part of the camouflaged object.

### 3.2 Overall Architecture

Our proposed CAReFuseNet, as illustrated in Figure 2, follows a dual-branch architecture. The first branch extracts the salient target object features, denoted as $E$, from $K$ reference images, $\{I_k^{ref}\}_{k=1}^K$, through the Ref-Image Encoder followed by the Ref-Feature Combiner. The second branch extracts multi-scale features, denoted as $\{F_j\}_{j=1}^4$, from the camouflaged image, $I^{camo}$, using the Camo-Image Encoder. Our novel cross-attention based reference feature fusion module, dubbed CAReFuse, fuses the reference features, $E$, with the image features, $\{F_j\}_{j=1}^4$, to form the reference-conditioned feature representations of the input camouflaged image. This process helps to better align the features of the camouflaged image with the referring salient object. These strongly aligned image feature representations are subsequently used by the segmentation decoder to generate a binary segmentation mask for the target camouflaged objects as specified by the referring image.

**Ref-Image Encoder.** Following Zhang et al. (2025), we adopt the encoder from pre-trained ICON (Zhuge et al., 2023) model (with Pyramid Vision Transformer (Wang et al., 2022) backbone as default) as Ref-

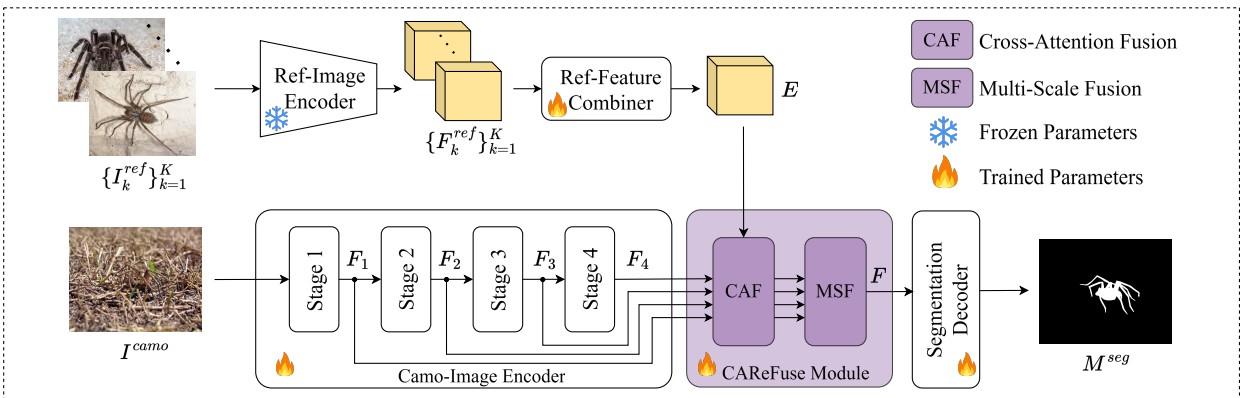

Figure 2: Overall architecture of our **CAReFuseNet** framework (Section 3.2). **Ref-Image Encoder** extracts features from reference images containing salient target objects. **Ref-Feature Combiner** combines individual reference image features to form a common feature representation, $E$, of the target object. **Camo-Image Encoder** extracts multi-scale features from input camouflaged image $I^{camo}$. **CAReFuse Module** fuses feature representations of salient target object with multi-scale features of $I^{camo}$. **Segmentation Decoder** generates a segmentation mask using the reference-conditioned feature representations.

Image Encoder to extract the respective feature representations $\{F_k^{ref}\}_{k=1}^K$, where $F_k^{ref} \in \mathbb{R}^{c_r \times h \times w}$, from the reference images $\{I_k^{ref}\}_{k=1}^K$, where $I_k^{ref} \in \mathbb{R}^{3 \times H \times W}$ (see Figure 2). Here, $h \times w$ represents the spatial dimension of the reference feature volume, while $c_r$ is its channel dimension. These individual reference feature representations are subsequently combined by Ref-Feature Combiner.

**Ref-Feature Combiner.** Ref-Feature Combiner combines $K$ reference feature representations, $\{F_k^{ref}\}_{k=1}^K$, where $F_k^{ref} \in \mathbb{R}^{c_r \times h \times w}$, into a single common salient target object representation, $E \in \mathbb{R}^{c_r \times h \times w}$ (see Figure 2), using spatial attention, channel attention followed by channel-wise average. Here, each feature volume comprises $c_r$ feature planes of $h \times w$ spatial dimension each. Finally, $E$ is computed as formulated below:

$$
\begin{aligned}
\tilde{F}_k^{ref} &= M_c(F_k^{ref}) \odot F_k^{ref}, \\
\hat{F}_k^{ref} &= M_s(\tilde{F}_k^{ref}) \odot \tilde{F}_k^{ref}, \\
E[c] &= \frac{1}{K} \sum_{k=1}^K \hat{F}_k^{ref}[c], \text{ for } c = 1, 2, ..., c_r
\end{aligned}
\tag{1}
$$

$E[c]$ is the $c^{th}$ feature plane of the combined reference feature volume $E$, and $\hat{F}_k^{ref}[c]$ is the $c^{th}$ feature plane from the $k^{th}$ feature volume $\hat{F}_k^{ref}$.

$$
M_c(.) = \sigma\Big(MLP\big(AvgPool_s(.)\big) + MLP\big(MaxPool_s(.)\big)\Big)
\tag{2}
$$

$$
M_s(.) = \sigma\Big(\mathcal{F}_{conv7 \times 7}\big([AvgPool_c(.); MaxPool_c(.)]\big)\Big)
\tag{3}
$$

**Camo-Image Encoder.** To extract multi-scale features from $I^{camo} \in \mathbb{R}^{3 \times H \times W}$, the pre-trained Pyramid Vision Transformer (PVT) (Wang et al., 2022) is chosen as the default Camo-Image Encoder. The features $\{F_j\}_{j=1}^4$, where $F_j \in \mathbb{R}^{c_j \times \frac{H}{2^{j+1}} \times \frac{W}{2^{j+1}}}$, are extracted from stage 1, stage 2, stage 3 and stage 4 of PVT, respectively (see Figure 2). The varying channel dimension, $c_j$, of the multi-scale features is adjusted to a common channel dimension, $c_d$, through the convolution operation. Subsequently, these channel-adjusted multi-scale features $\{F_j\}_{j=1}^4$, where $F_j \in \mathbb{R}^{c_d \times \frac{H}{2^{j+1}} \times \frac{W}{2^{j+1}}}$ are passed to the CAReFuse module for feature fusion. Importantly, all downstream fusion in CAReFuse module (cross-attention and multi-scale fusion) operates on this unified $c_d$-dimensional space. This contrasts with prior work such as UAT, which performs

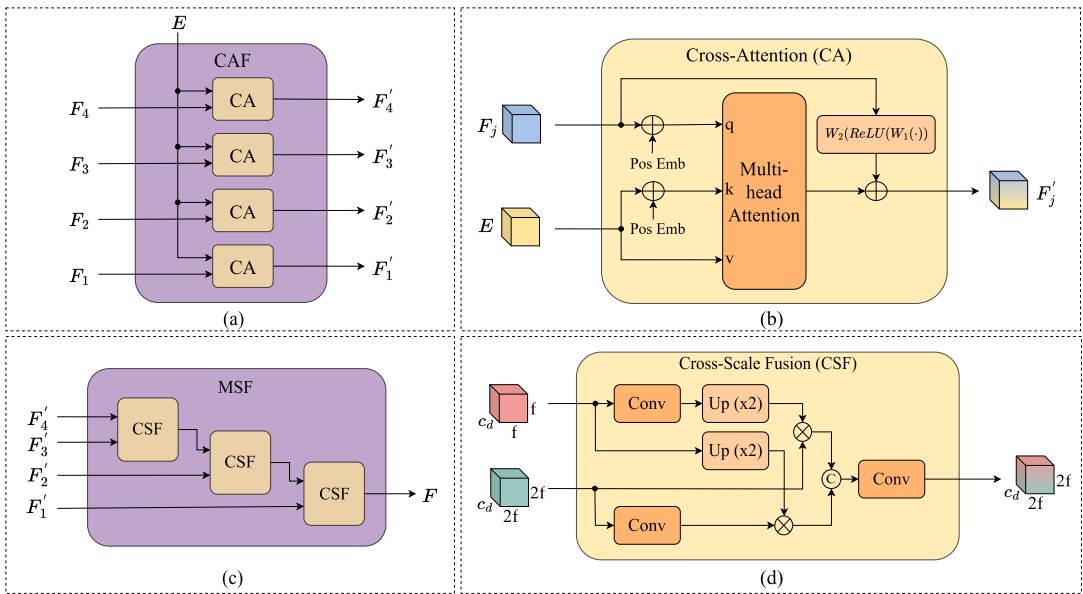

Figure 3: **CAReFuse Module** (Section 3.3): cross-attention based feature fusion module that fuses reference image features with multi-scale camouflaged image features. (a) **Cross-Attention Fusion (CAF) Module**: fuses reference image features $E$ with multi-scale camouflaged image features $F_1, F_2, F_3$ and $F_4$ individually. (b) **Cross-Attention (CA) Module**: performs multi-head cross-attention using camouflaged image features for *queries* and reference image features for *keys* and *values*. (c) **Multi-Scale Fusion (MSF) Module**: fuses reference-conditioned multi-scale features of camouflaged image into a single feature volume through cross-scale fusion. (d) **Cross-Scale Fusion (CSF) Module:** fuses a feature volume of lower spatial dimension with a feature volume of higher spatial dimension through convolution and up-sampling operations. $\oplus$ - element-wise sum, $\otimes$ - element-wise multiplication, and $\copyright$ - concatenate operation.

cross-attention and probabilistic decoding in a substantially higher-dimensional feature space, leading to a much larger number of trainable parameters.

CAReFuse module, described in detail in Section 3.3, performs feature fusion between $E$ and $\{F_j\}_{j=1}^4$ using cross-attention fusion followed by multi-scale fusion. The final feature representations produced by the CAReFuse module are denoted as $F \in \mathbb{R}^{c_d \times \frac{H}{4} \times \frac{W}{4}}$.

**Segmentation Decoder.** Similar to R2CNet (Zhang et al., 2025), we have used a convolution head as the segmentation decoder. The convolution head maps the final reference-conditioned feature representations $F$ to the segmentation mask $M^{seg} \in \{0, 1\}^{1 \times H \times W}$ through a series of convolution blocks.

### 3.3   CAReFuse Module

To segment the camouflaged objects of category $c$ present in the image $I^{camo}$, using reference images $\{I_k^{ref}\}_{k=1}^K$ containing salient examples of object category $c$, reference-conditioned feature representations of $I^{camo}$ must be extracted. To achieve this effectively through reference feature fusion, we propose a novel feature fusion module based on cross-attention. We denote these reference-conditioned camouflaged image features as $F$.

Our proposed CAReFuse module contains two submodules, namely, the Cross-Attention Fusion (CAF) and the Multi-Scale Fusion (MSF) modules.

The multi-scale features of the camouflaged image, $\{F_j\}_{j=1}^4$, are extracted at different spatial scales. CAF module fuses the referring salient target object features, $E$, with $\{F_j\}_{j=1}^4$ individually to form $\{F_j'\}_{j=1}^4$

respectively. This fusion operation can be formulated as:

$$\{F_j^{'}\}_{j=1}^4 = \text{CAF}(\{F_j\}_{j=1}^4, E).$$ (4)

To integrate information from different spatial scales, the reference-fused multi-scale features, $\{F_j^{'}\}_{j=1}^4$, are fused together using the MSF module. The resultant feature representations, $F$, can be expressed as:

$$F = \text{MSF}(\{F_j^{'}\}_{j=1}^4).$$ (5)

### 3.3.1 Cross-Attention Fusion Module

The Cross-Attention Fusion module contains four Cross-Attention (CA) blocks, each of which fuses $E$ with $\{F_j\}_{j=1}^4$ respectively. Figure 3(a) illustrates this process.

**Cross-Attention Block.** This block is illustrated in Figure 3(b). The reference feature volume $E \in \mathbb{R}^{c_r \times h \times w}$ is reshaped as a sequence of tokens denoted as $\phi_E \in \mathbb{R}^{hw \times c_r}$. Similarly, the feature volume $F_j \in \mathbb{R}^{c_d \times \frac{H}{2^{j+1}} \times \frac{W}{2^{j+1}}}$, extracted from the camouflaged image, is also reshaped as a sequence of tokens denoted as $\phi_{F_j} \in \mathbb{R}^{(\frac{H}{2^{j+1}} \frac{W}{2^{j+1}}) \times c_d}$. Now, multi-headed cross-attention (Vaswani et al., 2017) is performed between the two sequences, using $\phi_{F_j}$ as *queries* and $\phi_E$ as *keys* and *values*, as formulated in equation 6. To induce positional information, we also add learnable positional embeddings to query and key tokens before computing the multi-headed attention.

$$\phi_{F_j^{'}} = \mathcal{F}_{softmax}\left(\frac{(\phi_{F_j}W^Q)(\phi_E W^K)^T}{\sqrt{d}}\right)\phi_E W^V$$ (6)

Here, $d$ is the dimension of the projected query and key vectors, while $W^Q \in \mathbb{R}^{c_d \times d}$, $W^K \in \mathbb{R}^{c_r \times d}$ and $W^V \in \mathbb{R}^{c_r \times c_d}$ are the projection matrices for query, key, and value vectors, respectively. For brevity, equation 6 shows the computation of cross-attention only with one attention head, though we use multiple attention heads in practice. The multi-headed cross-attention output, $\phi_{F_j^{'}} \in \mathbb{R}^{(\frac{H}{2^{j+1}} \frac{W}{2^{j+1}}) \times c_d}$, has the same dimension as that of the input $\phi_{F_j}$. Inspired by the cross-attention fusion process in Sketch-guided Vision Transformer Encoder (Tripathi et al., 2024), we nonlinearly project $\phi_{F_j}$ and add it to the output of the multi-head attention. This process can be formulated as:

$$\phi_{F_j^{'}} = W_2 \cdot \mathcal{F}_{ReLU}(W_1 \cdot \phi_{F_j}) + \phi_{F_j^{'}},$$ (7)

where, $W_1 \in \mathbb{R}^{d^{'} \times (\frac{H}{2^{j+1}} \frac{W}{2^{j+1}})}$ and $W_2 \in \mathbb{R}^{(\frac{H}{2^{j+1}} \frac{W}{2^{j+1}}) \times d^{'}}$ are the trainable projection matrices.

The output sequence of tokens $\phi_{F_j^{'}} \in \mathbb{R}^{(\frac{H}{2^{j+1}} \frac{W}{2^{j+1}}) \times c_d}$ are reshaped into the feature volume $F_j^{'} \in \mathbb{R}^{c_d \times \frac{H}{2^{j+1}} \times \frac{W}{2^{j+1}}}$. The reference-fused multi-scale features, $\{F_j^{'}\}_{j=1}^4$, are then fused together using the MSF module.

Note that at each scale, only a single cross-attention block is applied between the camouflaged image features and the reference features, followed by a lightweight non-linear projection. Together with the convolution-only multi-scale fusion (see Section 3.3.2) and segmentation decoder, this design keeps the fusion head compact while still enabling rich global interactions between camouflaged and reference features.

### 3.3.2 Multi-Scale Fusion Module

The Multi-Scale Fusion module progressively fuses the reference-fused multi-scale features, $\{F_j^{'}\}_{j=1}^4$, to form a single feature volume $F$, as illustrated in Figure 3(c). This is performed using three Cross-Scale Fusion (CSF) blocks, each of which fuses a feature volume of the lower spatial dimension with that of the higher spatial dimension, through convolution and up-sampling operations.

$$F = \text{CSF}(\text{CSF}(\text{CSF}(F_4^{'}, F_3^{'}), F_2^{'}), F_1^{'})$$ (8)

Table 1: Comparison of our CAReFuseNet with Ref-COD models. '-Ref': R2CNet's referring framework applied to the existing COD model (results taken from (Zhang et al., 2025)). 'Tr.Params': Number of trainable parameters in million. 'Tot.Params': Number of total parameters in million. 'Macs': Multiply-accumulate operations in billion. '↑': the higher the better. '↓': the lower the better. Settings: image size $352 \times 352$, number of reference images $K = 5$. For Ours, metrics are reported as $mean(\pm std)$ over three random seed runs. The best two values are shown in **bold** and underline, respectively. Results of R2CNet Zhang et al. (2025) and UAT Wu et al. (2025) are taken from the respective prior works.

| Model | Backbone | Tr.Params(M) | Tot.Params(M) | Macs(G) | $S_m \uparrow$ | $\alpha E \uparrow$ | $wF \uparrow$ | $M \downarrow$ |
|---|---|---|---|---|---|---|---|---|
| R2CNet | ResNet-50 | 27.15 | 60.23 | 23.23 | 0.805 | 0.879 | 0.669 | 0.036 |
| PFNet-Ref | ResNet-50 | 57.58 | 90.66 | 59.59 | 0.811 | 0.885 | 0.687 | 0.036 |
| PreyNet-Ref | ResNet-50 | 38.70 | 71.78 | 117.60 | 0.817 | 0.900 | 0.704 | 0.032 |
| DGNet-Ref | EfficientNet-B5 | 20.10 | 53.18 | 7.24 | 0.821 | 0.891 | 0.696 | 0.032 |
| SINetV2-Ref | Res2Net-50 | 27.70 | 60.78 | 26.01 | 0.823 | 0.888 | 0.700 | 0.033 |
| BSANet-Ref | Res2Net-50 | 33.07 | 66.15 | 66.08 | 0.830 | 0.912 | 0.727 | 0.030 |
| ZoomNet-Ref | Tripple ResNet-50 | 33.30 | 66.38 | 218.24 | 0.834 | 0.886 | 0.720 | 0.029 |
| BGNet-Ref | Res2Net-50 | 151.06 | 184.14 | 171.03 | 0.840 | 0.909 | 0.738 | 0.029 |
| UAT | PVT-v2 | 111.60 | 173.64 | 61.70 | 0.855 | 0.912 | 0.757 | 0.026 |
| Ours-R | ResNet-50 | 27.21 | 50.72 | 13.79 | 0.821($\pm$0.001) | 0.898($\pm$0.001) | 0.705($\pm$0.001) | 0.033($\pm$0.001) |
| Ours-E | EfficientNet-B5 | 31.65 | 55.16 | 11.34 | 0.830($\pm$0.001) | 0.906($\pm$0.001) | 0.720($\pm$0.001) | 0.029($\pm$0.000) |
| Ours-S | Swin-Tiny | 21.93 | 108.61 | 29.30 | 0.842($\pm$0.001) | 0.912($\pm$0.001) | 0.738($\pm$0.001) | 0.028($\pm$0.001) |
| Ours-C | ConvNeXt-Tiny | 30.89 | 54.40 | 13.91 | 0.848($\pm$0.001) | 0.919($\pm$0.001) | 0.751($\pm$0.001) | 0.027($\pm$0.001) |
| Ours-P | PVT-v2 | 27.22 | 89.30 | 20.15 | **0.861**($\pm$0.001) | **0.927**($\pm$0.001) | **0.776**($\pm$0.001) | **0.024**($\pm$0.000) |

Note: $std$ is rounded to three decimal points. Values below 0.0005 are rounded to 0.000.

**Cross-Scale Fusion Block.** The process of cross-scale fusion, inspired by Zheng et al. (2023), is illustrated in Figure 3(d). The multi-scale features, $\{F'_j\}^4_{j=1}$, have their spatial dimensions halved for $j = 1$ to 4. Therefore, the fusion between the feature volumes of two different spatial scales, say, $U \in \mathbb{R}^{c_d \times f \times f}$ and $V \in \mathbb{R}^{c_d \times 2f \times 2f}$, is formulated as:

$$
\begin{aligned}
u &= \mathcal{F}_{up}(\mathcal{F}_{conv1\times1}(U)) \odot V, \\
v &= \mathcal{F}_{up}(U) \odot \mathcal{F}_{conv1\times1}(V), \\
\text{CSF}(U, V) &= \mathcal{F}_{conv1\times1}(\mathcal{F}_{concate}(u, v)),
\end{aligned}
\tag{9}
$$

where, $\mathcal{F}_{up}(.)$ indicates up-sampling, $\mathcal{F}_{conv1\times1}(.)$ is a $1 \times 1$ convolution operation, and $\otimes$ represents element-wise multiplication. Here, the up-sampling operation is performed to match the spatial dimensions.

# 4 Experiments and Results

We explain the training and evaluation setup in Section 4.1. Next, we report the results of the quantitative evaluation and the ablation studies in Section 4.2 and Section 4.3, respectively. Finally, Section 4.4 discusses results of qualitative evaluation.

## 4.1 Training and Evaluation Setup

Following the existing Ref-COD framework, R2CNet (Zhang et al., 2025), we choose structure loss (Wei et al., 2020) as our training objective. This function is formulated as:

$$
\mathcal{L}(P, G) = \sum_{i=1}^{4} \mathcal{L}_{bce}(P_i, G) + \mathcal{L}_{iou}(P_i, G),
\tag{10}
$$

where $P = \{M^{seg}, M_4^{scale}, M_3^{scale}, M_2^{scale}\}$ refers to the predicted segmentation masks and $G$ represents the ground truth. Here, $M^{seg}$ is the final predicted segmentation mask, while $M_4^{scale}, M_3^{scale}, M_2^{scale}$ are the mask predictions using the multi-scale features at the output of the Cross-Attention Fusion module.

**Implementation Details.** We adopt Adam optimizer (Kingma & Ba, 2015) to train our network for 100 epochs with a batch size of 32. We choose the initial learning rate of 5e-4 that eventually decays according

Table 2: Comparison of our CAReFuseNet (with different backbone choices) with CIRCOD (under Ref-COD setting). Settings: image size $512 \times 512$, number of ref images $K = 1$. The best two values are shown in **bold** and underline, respectively. The results of CIRCOD are taken from the prior work Gupta et al. (2025).

| Model | Backbone | $S_m \uparrow$ | $\alpha E \uparrow$ | $wF \uparrow$ | $M \downarrow$ |
|---|---|---|---|---|---|
| CIRCOD (Gupta et al., 2025) | PVT-v2 | 0.848 | 0.918 | 0.756 | 0.026 |
| Ours-R | ResNet-50 | 0.837 | 0.906 | 0.732 | 0.030 |
| Ours-E | EfficientNet-B5 | 0.859 | 0.926 | 0.773 | 0.024 |
| Ours-S | Swin-Tiny | 0.861 | 0.925 | 0.773 | 0.025 |
| Ours-C | ConvNeXt-Tiny | 0.861 | 0.929 | 0.778 | 0.024 |
| Ours-P | PVT-v2 | **0.880** | **0.940** | **0.815** | **0.021** |

to cosine annealing (Loshchilov & Hutter, 2017). During training, the Ref-Image Encoder parameters are frozen while the rest of the network parameters are updated. The parameters of the pre-trained Camo-Image Encoder are updated using 0.1 times the original learning rate used for updating the rest of the trainable parameters. During both training and evaluation, we resize the images to the default size of $352 \times 352$. The experiments are implemented using PyTorch (Paszke et al., 2019) framework and run on NVIDIA RTX 6000 Ada GPU.

**Evaluation Metrics.** To evaluate our model, we adopt the four metrics commonly used in foreground map evaluation, including structure-measure ($S_m$) (Fan et al., 2017), adaptive E-measure ($\alpha E$) (Fan et al., 2018), weighted F-measure ($\omega F$) (Margolin et al., 2014) and mean absolute error (M) (Perazzi et al., 2012).

## 4.2 Quantitative Evaluation

We compare the performance of our CAReFuseNet with CIRCOD (Gupta et al., 2025) and with remaining Ref-COD methods separately, since CIRCOD uses different settings. Table 1 compares our CAReFuseNet with R2CNet (and all other Ref-COD extended models using R2CNet's framework) (Zhang et al., 2025) and UAT (Wu et al., 2025), which use image size $352 \times 352$ and the number of reference images $K = 5$. Table 2 compares with CIRCOD that uses image size $512 \times 512$ and $K = 1$. To verify the generality of our method, we evaluated our CAReFuseNet with several other backbone architectures in addition to Pyramid Vision Transformer (PVT) Wang et al. (2022), including Swin-Tiny Transformer (Sw-T) (Liu et al., 2021), ResNet-50 (R-50) (He et al., 2016), ConvNeXt-Tiny (Cxt-T) (Liu et al., 2022) and EfficientNet-B5 (E-B5) (Tan & Le, 2019). When using Sw-T backbone for Camo-Image Encoder, we use ICON (Zhuge et al., 2023) with Swin backbone as Ref-Image Encoder. Whereas, when using R-50, E-B5, Cxt-T for Camo-Image Encoder, we use ICON with ResNet-50 backbone as Ref-Image Encoder.

As can be observed in Table 1, with only 27% as many trainable parameters and 53% as many total inference parameters, our method (Ours-P) outperforms UAT. This reduction arises from several architectural choices. First, CAReFuseNet uses the ICON encoder as a frozen Ref-Image Encoder, so a large fraction of the total parameters do not contribute to the trainable budget. Second, all multi-scale camouflaged features are projected to a unified channel dimension $c_d = 256$, and both the Cross-Attention Fusion and Multi-Scale Fusion modules operate at this reduced dimensionality. Third, reference–image fusion is implemented with a single cross-attention block per scale followed by shallow $1 \times 1$ convolutions for cross-scale fusion and a lightweight convolutional decoder. Table 2 shows that our method also outperforms CIRCOD in all metrics under E-B5, Sw-T, CxT-T and PVT backbones.

## 4.3 Ablative Studies

**Ablation on $c_d$.** As the channel dimension of the multi-scale camouflaged image features (*i.e.* $c_d$) impacts the number of parameters of the model, we conduct an ablation study on $c_d$ to assess its effect on performance. We vary the channel dimension from 64 to 512 in powers of two, training and evaluating CAReFuseNet for each configuration. As shown in Table 3, an increase in $c_d$ from 64 to 256 improves performance, while the performance gain saturates with a further increase. Therefore, for a better trade-off between performance and computational efficiency, and to avoid over-parameterization, we select 256 as the default

**Table 3:** Ablation on channel dimension (*i.e.* $c_d$) of our CAReFuseNet with different backbone choices. 'Param': Number of model parameters in million. 256 is chosen as the default channel dimension, striking a balance between performance and computational efficiency.

| $c_d$ | CAReFuseNet-R (Ours-R) | | | | CAReFuseNet-E (Ours-E) | | | | CAReFuseNet-S (Ours-S) | | | | CAReFuseNet-C (Ours-C) | | | | CAReFuseNet-P (Ours-P) | | | |
|---|---|---|---|---|---|---|---|---|---|---|---|---|---|---|---|---|---|---|---|---|
| | $S_m\uparrow$ | $\alpha E\uparrow$ | $wF\uparrow$ | $M\downarrow$ | $S_m\uparrow$ | $\alpha E\uparrow$ | $wF\uparrow$ | $M\downarrow$ | $S_m\uparrow$ | $\alpha E\uparrow$ | $wF\uparrow$ | $M\downarrow$ | $S_m\uparrow$ | $\alpha E\uparrow$ | $wF\uparrow$ | $M\downarrow$ | $S_m\uparrow$ | $\alpha E\uparrow$ | $wF\uparrow$ | $M\downarrow$ |
| 64 | 0.820 | 0.895 | 0.699 | 0.033 | 0.825 | 0.898 | 0.711 | 0.030 | 0.840 | 0.907 | 0.732 | 0.029 | 0.845 | 0.916 | 0.745 | **0.027** | 0.858 | 0.921 | 0.768 | 0.025 |
| 128 | 0.822 | 0.897 | 0.704 | 0.033 | 0.829 | 0.904 | 0.718 | 0.029 | **0.843** | **0.915** | **0.740** | **0.028** | 0.846 | 0.916 | 0.748 | 0.027 | 0.859 | 0.925 | 0.773 | **0.024** |
| 256 | **0.821** | **0.898** | **0.705** | 0.033 | **0.830** | **0.906** | **0.720** | **0.029** | 0.842 | 0.912 | 0.738 | **0.028** | **0.848** | **0.919** | **0.751** | **0.027** | **0.861** | 0.927 | 0.776 | **0.024** |
| 512 | **0.821** | **0.899** | **0.705** | **0.032** | 0.829 | **0.906** | **0.720** | **0.029** | **0.843** | 0.913 | 0.738 | **0.028** | 0.847 | 0.918 | **0.751** | **0.027** | 0.860 | **0.928** | **0.777** | **0.024** |

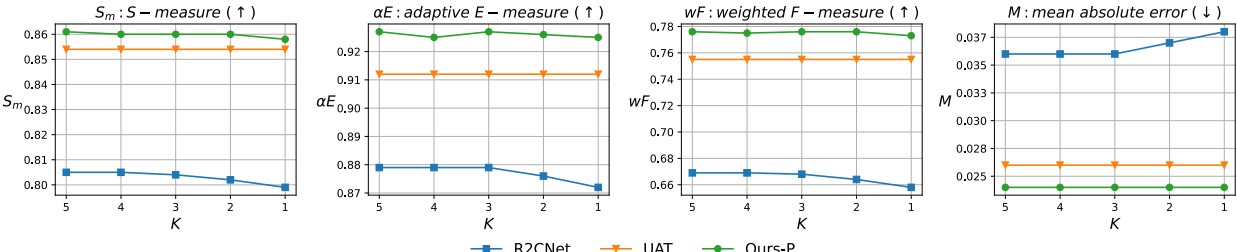

**Figure 4:** Comparison among Ours-P, UAT and R2CNet w.r.t ablation experiment on number of reference images (*i.e.*, $K$), where $K$ is varied consistently between training and test.

channel dimension. This design, together with the parameter-efficient fusion and decoding modules, also contributes to the substantial parameter savings observed in Table 1 when comparing Ours-P to UAT, which relies on higher-dimensional transformer blocks for both reference feature fusion and decoding.

**Ablation on $K$.** We trained and evaluated our CAReFuseNet by varying the number of reference images (variation in $K$ is same during training and test time), $5 \geq K \geq 1$, used to guide camouflaged object detection. Our results are compared with those of UAT and R2CNet in Figure 4. The comparison demonstrates that our CAReFuseNet outperforms both UAT and R2CNet, even when utilizing only one reference image, while others using five. Additionally, we also evaluated our method in an alternative setting where we train once with $K = 5$ and evaluate at different $K \in \{5, 4, 3, 2, 1\}$, the results of which are included in Appendix D.

**Ablation on Ref-Feature Combiner.** As formulated in Equations 1, 2 and 3, our Ref-Feature Combiner performs channel attention (CA) followed by spatial attention (SA) on the individual reference feature volumes, $\{F_k^{ref}\}_{k=1}^{K} \in \mathbb{R}^{c_r \times h \times w}$, extracted by Ref-Image Encoder. Subsequently, a learnable channel-wise weighted sum is performed on these $K$ feature volumes. We conduct an ablation study on various operations involved in the Ref-Feature Combiner. As shown in Table 4, our chosen strategy gives better results among all choices.

**Ablation on CAReFuse Module Components.** To analyze the importance of each component of the CAReFuse module, we conducted ablation studies, with results reported in Table 5. While the CAF module, when used alone, yields improvements of 0.1%, 0.3% in metrics $S_m$, $\alpha E$, the MSF module delivers gains of

**Table 4:** Ablation study on the Ref-Feat Combiner module. 'CA': Channel Attention, 'SA': Spatial Attention, 'WS': Weighted Sum, 'CWA': Channel-wise Average. Metrics are reported as $mean(\pm std)$ over three random seed runs.

| CA | SA | CWA | $S_m\uparrow$ | $\alpha E\uparrow$ | $wF\uparrow$ | $M\downarrow$ |
|---|---|---|---|---|---|---|
| ✗ | ✗ | ✓ | 0.859(±0.001) | 0.926(±0.001) | 0.775(±0.001) | 0.024(±0.000) |
| ✗ | ✓ | ✓ | 0.860(±0.001) | 0.925(±0.002) | 0.775(±0.002) | 0.024(±0.000) |
| ✓ | ✗ | ✓ | 0.859(±0.001) | 0.924(±0.002) | 0.773(±0.003) | 0.024(±0.000) |
| ✓ | ✓ | ✓ | 0.861(±0.001) | 0.927(±0.001) | 0.776(±0.001) | 0.024(±0.000) |

Note: *std* is rounded to three decimal points.

**Table 5:** Ablation study on the components of the CAReFuse module. 'CAF': Cross Attention Fusion, 'MSF': Multi Scale Fusion. Metrics are reported as $mean(\pm std)$ over three random seed runs.

| CAF | MSF | $S_m\uparrow$ | $\alpha E\uparrow$ | $wF\uparrow$ | $M\downarrow$ |
|---|---|---|---|---|---|
| ✗ | ✗ | 0.807(±0.001) | 0.855(±0.003) | 0.654(±0.001) | 0.038(±0.000) |
| ✓ | ✗ | 0.808(±0.001) | 0.858(±0.002) | 0.654(±0.003) | 0.038(±0.001) |
| ✗ | ✓ | 0.858(±0.001) | 0.924(±0.001) | 0.773(±0.002) | 0.025(±0.001) |
| ✓ | ✓ | 0.861(±0.001) | 0.927(±0.001) | 0.776(±0.001) | 0.024(±0.000) |

Note: *std* is rounded to three decimal points.

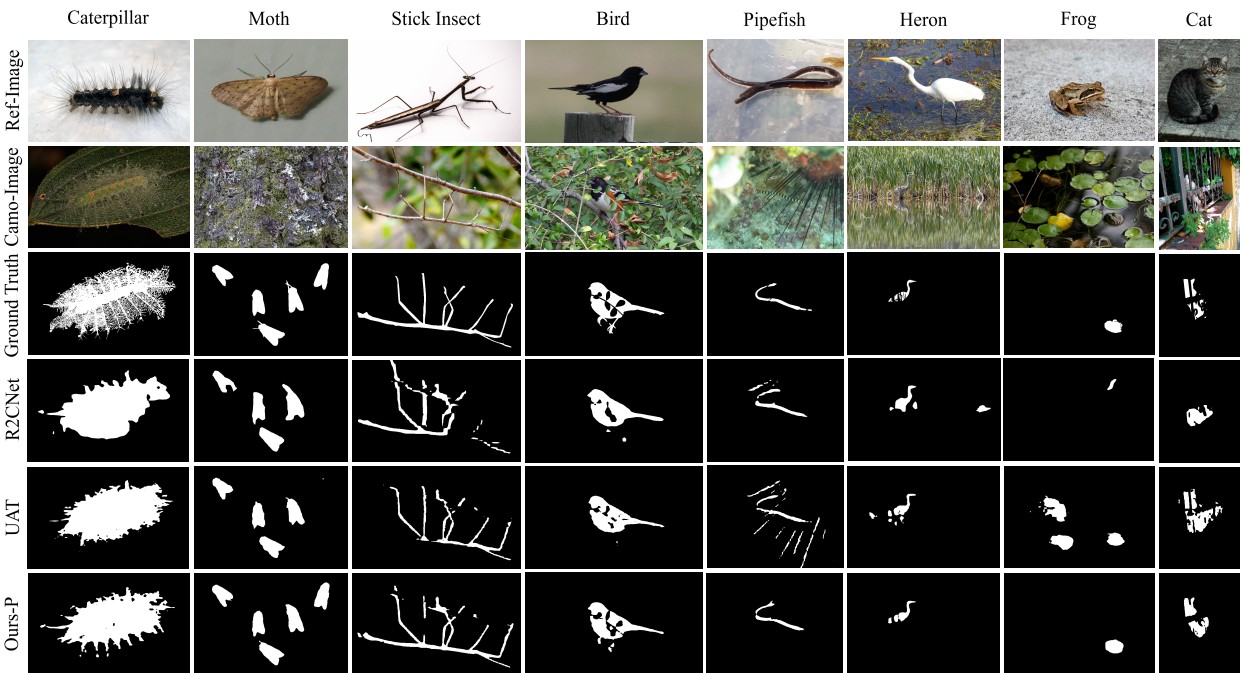

Figure 5: Visual comparison between the predictions of a state-of-the-art Ref-COD methods (*i.e.* R2CNet, UAT) and our CAReFuseNet-P. Our method evidently outperforms state-of-the-art.

5.1%, 6.9%, 11.9% and 1.3%, in metrics $S_m$, $\alpha E$, $\omega F$ and $M$, respectively. When combined, the two modules together achieve gains of 5.4%, 7.2%, 12.2% and 1.4% in the respective metrics.

## 4.4 Qualitative Evaluation

We present a qualitative comparison of the mask predictions of CAReFuseNet (Ours-P) with other Ref-COD methods in Figure 5. As can be observed from the visualization, the predictions of our method are substantially superior compared to those of the other methods in highly challenging scenarios such as very small or narrow objects, occluded objects, and objects with complex boundaries. As can be noticed in case of *Moth*, *Heron* and *Frog*, R2CNet and UAT either miss some smaller objects or detect false positives in the camouflaged images, while our method segments them accurately. In case of *Stick Insect* and *Pipefish*, it can be observed that our method detects very narrow parts of the objects more effectively, while other methods fail. The case of *Bird* and *Cat* proves the effectiveness of our CAReFuseNet in detecting and accurately segmenting objects under severe occlusion, where R2CNet and UAT struggle. The predicted masks for *Caterpillar* demonstrate that our CAReFuseNet segmented the objects with complex boundaries more precisely than other methods. Another notable strength of our method is its superior

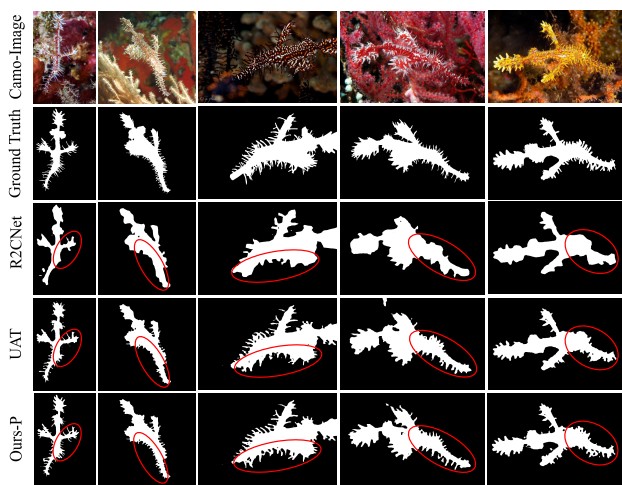

Figure 6: Visual comparision between the masks produced by R2CNet, UAT and Ours-P for camouflaged objects with high frequency edges. Our method produces superior quality masks, especially evidenced by the finer details being better captured.

ability to generate high-quality segmentation masks for camouflaged objects, particularly in regions with high-frequency edges. As illustrated in Figure 6, the segmentation masks produced by our CAReFuseNet capture object boundaries with greater accuracy and preserve fine edge details more effectively than those generated by R2CNet. Remarkably, with significantly less number of trainable and total inference parameters (see Table 1), our method still narrowly outperforms UAT in producing high quality edges. This qualitative comparison clearly demonstrates the effectiveness of our CAReFuseNet (Ours-P) in handling challenging edge-rich regions, underscoring its overall superiority in segmenting camouflaged objects.

## 5 Conclusion

Given the difficult nature of the COD problem in general and the practical utility of the Ref-COD task in particular, this paper proposes a novel cross-attention based reference feature fusion network (CAReFuseNet) for segmenting camouflaged objects in an image under the guidance of reference images containing the salient target object. The method introduces a cross-attention based feature fusion module (CAReFuse module) to extract the reference-conditioned feature representations of camouflaged images, combining a Cross-Attention Fusion (CAF) submodule with a Multi-Scale Fusion (MSF) submodule that operates in a unified low-dimensional feature space. By freezing the Ref-Image Encoder, projecting all camouflaged image features to a shared reduced channel dimension, and employing a single cross-attention block per scale followed by lightweight convolutional fusion and decoding, the overall architecture is explicitly designed to be parameter-efficient.

Extensive experiments on the R2C7K dataset demonstrate that CAReFuseNet achieves strong quantitative performance while being substantially more compact than prior approaches. In particular, the PVT-v2 instantiation (Ours-P) surpasses Uncertainty-Aware Transformer (UAT), despite using only about one-quarter of UAT's trainable parameters and roughly half of the total inference parameters (Table 1). Under the alternative setting adopted by CIRCOD, CAReFuseNet also attains superior performance across multiple backbones (Table 2). Qualitative results further show that the proposed model yields marked improvements in challenging scenarios such as very small or narrow objects, heavily occluded objects, and objects with complex boundaries, while still preserving fine edge details.

Overall, this work highlights that explicit attention to parameter efficiency in the design of reference feature fusion can yield Ref-COD models that are both accurate and lightweight. The proposed CAReFuseNet provides a practical, parameter-efficient baseline for future research on Ref-COD.

**Limitations and Future Directions**

Like other Ref-COD methods (e.g., R2CNet (Zhang et al., 2025) and UAT (Wu et al., 2025)), our method does not distinguish the scenarios where the camouflaged image to be segmented does not contain the target object specified by the reference images; it assumes the camouflaged image to always contain the specified target objects. It would be valuable to extend our parameter-efficient Ref-COD method to accommodate more generalized scenes without this restricted assumption. Furthermore, while accepting multiple reference images, our method tries to segment a single target object category (specified by the set of reference images) at a time. It will also be promising to extend the Ref-COD methods to accept multi-category references, enabling them to segment the target objects of multiple categories in the camouflaged image. A potential architectural pathway to achieve this involves a two-stage adaptation: first, the model must be adapted to output a null (blank) segmentation mask when presented with mismatched or negative reference images. Once this capability is achieved, multi-category support can be realized by instantiating parallel streams of our CAReFuse module - one for each reference category - to individually align the camouflaged image features with their respective reference features. These category-specific aligned features can then be processed through a shared segmentation decoder to yield distinct, parallel prediction masks. Finally, these individual predictions can be combined into a comprehensive multi-category segmentation mask.

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

# A   Additional Ablative Studies

In this section, we present the additional ablation studies on the supervision strategy followed, number of attention heads used in cross-attention fusion, and on segmentation decoder.

## A.1   Ablation on Supervision

Table 6: Ablation study on supervision strategy.

| No. | Supervision Strategy | $S_m \uparrow$ | $\alpha E \uparrow$ | $wF \uparrow$ | $M \downarrow$ |
|---|---|---|---|---|---|
| 1 | Single Supervision (Eq. 11) | 0.857($\pm$0.001) | 0.924($\pm$0.001) | 0.771($\pm$0.002) | 0.024($\pm$0.001) |
| 2 | Multi-Stage Supervision (Eq. 10) | 0.861($\pm$0.001) | 0.927($\pm$0.001) | 0.776($\pm$0.001) | 0.024($\pm$0.000) |

Note: *std* is rounded to three decimal points. Values below 0.0005 are rounded to 0.000.

Equation 10 shows our supervision strategy, where supervision is applied to mask predictions computed at multiple scales using multiscale features from multiple stages of the Camo-Image Encoder. We conducted an ablation study on the supervision to compare our chosen strategy with an alternative strategy, where supervision is applied only to the final segmentation mask as described below:

$$\mathcal{L}(P, G) = \mathcal{L}_{bce}(P, G) + \mathcal{L}_{iou}(P, G), \tag{11}$$

where $P$ refers to the predicted segmentation mask $M^{seg}$, and $G$ is the ground truth mask. Table 6 shows that our chosen supervision strategy yields a marginal improvement over the alternative strategy.

## A.2 Ablation on Number of Attention Heads

Table 7: Ablation study on number of attention heads in cross attention fusion.

| # Attention heads | $S_m \uparrow$ | $\alpha E \uparrow$ | $wF \uparrow$ | $M \downarrow$ |
|---|---|---|---|---|
| 1 | 0.860($\pm$0.001) | 0.925($\pm$0.001) | 0.776($\pm$0.002) | 0.024($\pm$0.000) |
| 4 | 0.861($\pm$0.001) | 0.927($\pm$0.001) | 0.776($\pm$0.001) | 0.024($\pm$0.000) |
| 8 | 0.859($\pm$0.001) | 0.926($\pm$0.001) | 0.775($\pm$0.001) | 0.025($\pm$0.001) |

Note: $std$ is rounded to three decimal points. Values below 0.0005 are rounded to 0.000.

We ablate on the number of attention heads used for the cross-attention fusion (Eq. 6), and the results are reported in Table 7.

## A.3 Ablation on Segmentation Decoder

To strengthen the claim that the performance gain of our method over UAT can be attributed to the cross-attention fusion module - CAReFuse, and to decouple the contributions of cross-attention fusion design from the channel-dimension and decoder choice, we ablate on the segmentation decoder. We chose channel dimension 512 along with a transformer decoder in place of the convolution-based decoder (closer to UAT design) and reported the ablation results in table 8. Ours-P, under both convolution-based and transformer-based decoder choices, outperforms UAT, attributing the performance gain of Ours-P (with the default convolution-based decoder) equally to the CAReFuse module rather than to the convolution decoder alone.

Table 8: Ablation study on segmentation decoder.

| | Decoder type | $S_m \uparrow$ | $\alpha E \uparrow$ | $wF \uparrow$ | $M \downarrow$ |
|---|---|---|---|---|---|
| UAT | Transformer Decoder | 0.855 | 0.912 | 0.757 | 0.026 |
| Ours-P | Transformer Decoder | 0.857($\pm$0.001) | 0.926($\pm$0.001) | 0.770($\pm$0.001) | 0.025($\pm$0.001) |
| | Convolution Decoder | 0.861($\pm$0.001) | 0.927($\pm$0.001) | 0.776($\pm$0.001) | 0.024($\pm$0.000) |

Note: $std$ is rounded to three decimal points. Values below 0.0005 are rounded to 0.000.

# B Speed and Memory Comparison

We compare Ours-P with R2CNet and UAT in terms of the speed (in Frames per second) and the GPU memory in (mega bytes). The comparison is reported in Table 9. In Table 10, we report inference speed and memory of Ours-P for different values of $K$. Inference speeds were measured as the average over 1000 iterations on NVIDIA RTX A5000.

Table 9: Speed and memory comparison.

| | Speed(FPS) | Memory(MB) | $S_m \uparrow$ | $\alpha E \uparrow$ | $wF \uparrow$ | $M \downarrow$ |
|---|---|---|---|---|---|---|
| R2CNet | 90.68 | 2155.75 | 0.805 | 0.879 | 0.669 | 0.036 |
| UAT | 33.62 | 3311.75 | 0.855 | 0.912 | 0.757 | 0.026 |
| Ours-P | 71.84 | 2043.75 | 0.861 | 0.927 | 0.776 | 0.024 |

Table 10: Speed and memory at different $K$.

| # Ref. Images($K$) | Speed(FPS) | Memory(MB) |
|---|---|---|
| $K = 5$ | 71.84 | 2043.75 |
| $K = 4$ | 73.77 | 2043.75 |
| $K = 3$ | 75.22 | 2043.75 |
| $K = 2$ | 76.59 | 2043.75 |
| $K = 1$ | 78.09 | 2043.75 |

## C  Analysis of Computation Cost vs Input Resolutions

In this section, we analyze how the computational overhead increases when scaling to higher input image resolutions. As discussed in Section 3.3.1, the cross-attention operation (Eq. 6) is performed between reference image features and camouflaged image features by reshaping those feature volumes as token sequences $\phi_E \in \mathbb{R}^{hw \times c_r}$ and $\phi_{F_j} \in \mathbb{R}^{(\frac{H}{2^{j+1}} \frac{W}{2^{j+1}}) \times c_d}$. An increase in the resolution of the input image (both $I^{ref}$ and $I^{camo}$) would result in an increase in the sequence length for $\phi_E$ and $\phi_{F_j}$. Denoting the sequence lengths as $M$ and $N$ for $\phi_E$ and $\phi_{F_j}$, respectively, and the query and key projection dimension as $d$, the theoretical computational complexity is given as below:

$$\mathcal{O}\Big(\underbrace{Nc_dd + Mc_rd + Mc_rc_d}_{\text{Projections}} + \underbrace{NMd}_{\text{Attention Matrix}} + \underbrace{NMc_d}_{\text{Value Aggregation}}\Big) \tag{12}$$

Consequently, the $\mathcal{O}(N.M(d+c_d))$ term constitutes the primary bottleneck, causing the computational cost to scale bilinearly with both sequence lengths. Specifically, if both sequences expand by a factor of $k$, the overall computational cost increases quadratically by a factor of $k^2$.

## D  Train Once and Evaluate at Different $K$

In addition to the ablation on $K$ in Section 4.3 (where we change $K$ by same value both during training and test time), we evaluated our model in 'train once and evaluate at different $K$' setting, by first training our model with $K = 5$ (5 reference images for each camouflaged image) and evaluating it at $K \in \{5, 4, 3, 2, 1\}$. The results are reported in Table 11.

Table 11: Results under *Train once and evaluate at different $K$* setting.

| # Ref. Images($K$) at test | $S_m \uparrow$ | $\alpha E \uparrow$ | $wF \uparrow$ | $M \downarrow$ |
|:---:|:---:|:---:|:---:|:---:|
| $K = 5$ | 0.861($\pm$0.001) | 0.927($\pm$0.001) | 0.776($\pm$0.001) | 0.024($\pm$0.000) |
| $K = 4$ | 0.860($\pm$0.000) | 0.926($\pm$0.001) | 0.776($\pm$0.001) | 0.024($\pm$0.000) |
| $K = 3$ | 0.860($\pm$0.000) | 0.926($\pm$0.001) | 0.775($\pm$0.000) | 0.024($\pm$0.000) |
| $K = 2$ | 0.860($\pm$0.000) | 0.927($\pm$0.001) | 0.775($\pm$0.000) | 0.024($\pm$0.000) |
| $K = 1$ | 0.859($\pm$0.000) | 0.927($\pm$0.001) | 0.774($\pm$0.000) | 0.025($\pm$0.001) |

Note: *std* is rounded to three decimal points. Values below 0.0005 are rounded to 0.000.

## E  Failure Cases

In this section, we discuss the example failure cases as shown in Figure 7, where Ours-P considerably fails to segment the camouflaged objects correctly. When the camouflaged object in the image is very small while significantly similar in texture to the surrounding objects (as in case of *SeaHorse* and *Lizard*) and when the camouflaged object is surrounded by background clutter having similar texture (as in *Cat*, *GhostPipeFish*, and *Duck*), our model fails by producing more false positive regions in the segmentation masks. In case of *Duck*, interestingly but incorrectly, our model produces a false positive prediction for the reflection of the duck in the water. In some cases where the texture of the camouflaged object is extremely similar to that of the background (as in case of *Fish*, *Gecko*, and *Flounder*), our model could segment the objects only partially, having significant false negative regions in the predicted segmentation masks.

## F  Predictions for Camouflaged Images with Negative Reference Images

Although the scope of the Ref-COD task addressed in this work is limited to the assumption that the given reference image always contains the target camouflaged object, in this section, we analyze how our model behaves when presented with the mismatched references, *i.e.,* the reference images not representing the target camouflaged object. Figure 8 presents four examples cases where the camouflaged images are presented with

mismatched (or negative) reference images. These four examples reveal that even with negative reference images, our model still incorrectly produces some false positive regions in the segmentation masks instead of producing a blank output. The other Ref-COD methods (UAV and R2CNet) also produced false positive regions when presented with negative references.

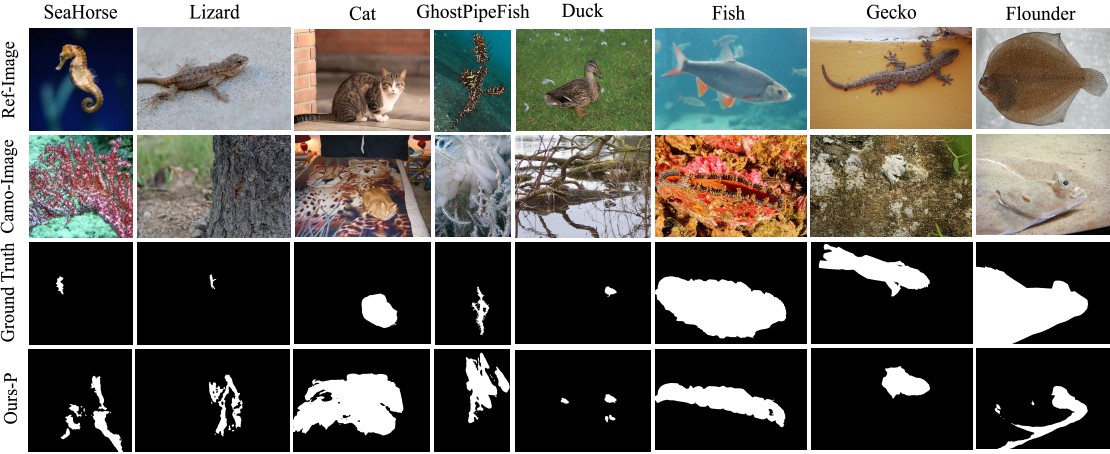

Figure 7: Examples of failure cases.

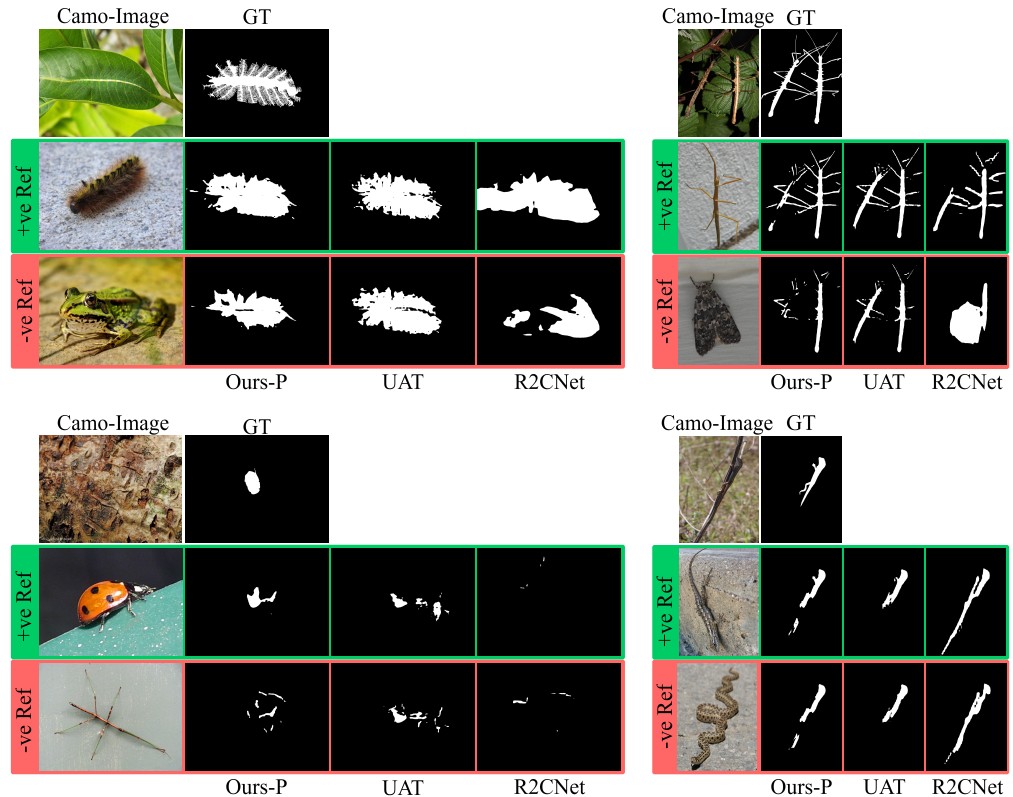

Figure 8: Qualitative analysis of predictions for negative reference images. **Top-left:** *Caterpillar* with *Frog* as a negative reference. **Top-right:** *StickInsect* with *Moth* as a negative reference. **Bottom-left:** *Bug* with *StickInsect* as a negative reference. **Bottom-right:** *Lizard* with *Snake* as a negative reference.

