# OpenReview forum: "CAReFuseNet: Cross Attention Fusion Network for Referring Camouflaged Object Detection"
_TMLR — Under review for TMLR_

### Review · Reviewer_AxL3 · 2026-05-06

**Summary Of Contributions:**

The paper proposes CAReFuseNet, a parameter-efficient framework for Referring Camouflaged Object Detection (Ref-COD). The core contribution is the CAReFuse module, composed of (1) a Cross-Attention Fusion (CAF) submodule that performs per-scale cross-attention between reference features $E$ and multi-scale camouflaged image features, and (2) a Multi-Scale Fusion (MSF) submodule that merges the reference-conditioned multi-scale features through cross-scale fusion blocks. All fusion is constrained to a unified low-dimensional space and paired with a frozen ICON-based Ref-Image Encoder and a lightweight convolutional decoder. Experiments on R2C7K show that the PVT-v2 instantiation surpasses UAT on standard Ref-COD metrics, with ~25% trainable and ~52% total inference parameters. Additional results are reported across ResNet-50, EfficientNet-B5, Swin-T, and ConvNeXt-T backbones.

**Additional Comments:**

**W1. Lack of statistical reliability of the performance gains.** Several of the central comparisons rely on small absolute differences. In Table 1, Ours-P vs UAT on structure-measure ($S_m$) is 0.861 vs 0.855 (+0.006), adaptive E-measure ($\alpha E$) 0.928 vs 0.912 (+0.016), and mean absolute error ($M$) 0.024 vs 0.026. In Table 4, the four Ref-Feat Combiner ablation rows differ by at most 0.002 on $S_m$ and 0.003 on $\alpha E$. None of these are reported with multiple seeds or standard deviations. The gap is so small that it must be supported by mean ± std over $\geq 3$ seeds before claims can be made.

**W2. Robustness to varying $K$ at inference.** The $K$ ablation in Figure 4 appears to vary $K$ consistently between training and evaluation. It would be interesting and practically relevant to also report a train-once / evaluate-at-different-$K$ setting (e.g., train at $K=5$, evaluate at $K \in \{1, \ldots, 4\}$). Note that the Ref-Feature Combiner uses a learned weight matrix $W \in \mathbb{R}^{K \times c_r}$ (Eq. 1), which is $K$-dependent by construction; a brief clarification on how $W$ would be handled at a different test-time $K$ (zero-padding, masking, re-normalization) would also help readers understand the model's flexibility.

**W3. The "lightweight decoder" / low-dim fusion attribution is claimed but not evaluated.** The paper claims parameter savings due to (a) frozen Ref-Image Encoder, (b) projection to $c_d = 256$, and (c) the lightweight convolutional decoder. These three factors are bundled in all the comparisons. It would strengthen the claim to ablate them: e.g., CAReFuse with $c_d = 512$ + transformer decoder (closer to UAT's design) vs the proposed configuration, on the same backbone. Otherwise, it is hard to tell whether the gain comes from the cross-attention fusion design or simply from replacing a transformer decoder with a conv decoder.

**Audience:**

Yes

**Audience Explanation:**

Ref-COD is an emerging task, and a parameter-efficient baseline that matches prior heavy transformer approaches with ~25% of trainable parameters is of practical interest to the referring-segmentation and resource-constrained vision communities.

**Claims And Evidence:**

No

**Claims Explanation:**

The parameter-efficiency claim is clearly supported by Tables 1–2, but the SOTA accuracy claim relies on small single-run gaps (e.g., +0.006 $S_m$ over UAT in Table 1; ≤ 0.003 differences across rows in Tables 4 and 6 and lacks mean ± std reporting.

**Requested Changes:**

1. **Multi-seed evaluation (W1).** Re-run the main comparison (Table 1) and the ablation studies (Tables 4, 5, 6) with at least 3 random seeds, and report mean ± std for all four metrics. This is especially important for Tables 4 and 6, where the current gaps (≤ 0.003 on $S_m$ / $\alpha E$) are within plausible seed-to-seed variance and cannot currently support the design conclusions drawn from them.

2. **Decoder and channel-dimension ablation (W3).** Add at least one configuration that decouples the contribution of the cross-attention fusion design from the decoder/channel-dimension choice. This would directly attribute the parameter savings and accuracy gain to the fusion design vs the decoder/dimensionality choice, which the current Tables 1–2 cannot distinguish.

---

> ### Author Response · Authors · 2026-05-29
> **Rebuttal by Authors**
>
> We thank the reviewer for the comments.
>
> **W1: Multi-seed evaluation.**
> **Response:** As recommended by the reviewer, we re-run the experiments of Tables 1, 4, 5, and 6 with 3 different random seeds and reported $mean \\pm std$. The tables are updated accordingly. Note: The standard deviation values below $0.0005$ are rounded to $0.000$.
>
> **W2: Robustness to varying K at inference.**
> **Response:** We acknowledge that our earlier formulation of $W\in{\\mathbb{R}^{K \\times c_r}}$ in Ref-Feat Combiner necessitates varying $K$ consistently between training and inference, since the $W$ is K-dependent and is fixed once the model is trained. To accommodate for the more flexible setting of *'train once and evaluate at different $K$'*, as suggested by the reviewer, we reformulated the Ref-Feat combiner using channel attention, spatial attention followed by channel-wise average. (Eq.1 in page 5 of the manuscript is updated.)
> This formulation would make the *'train once and evaluate at different $K$'* setting inherently possible. We re-run all the experiments using the modified formulation, and all the relevant tables and figures are updated accordingly. The performance trends of our method remain largely unchanged. Additionally, we reported the results of *'train once and evaluate at different $K$'* in Table 11 in Appendix D (page 18).
>
> **W3: Decoder/Channel-dimension ablation.**
> **Response:** To decouple the contribution of cross-attention fusion design from the decoder choice, as suggested by the reviewer, we conducted an ablation study on decoder by choosing a transformer-based decoder (closer to UAT design) in place of convolution-based decoder (our default choice). The results are reported in Table 8 of Appendix A3 (page 17).
> | | Decoder type | $S_m\uparrow$ | $\alpha E\uparrow$ | $wF\uparrow$ | $M\downarrow$ |
> | :--- | :--- | :---: | :---: | :---: | :---: |
> | UAT | Transformer Decoder | 0.855 | 0.912 | 0.757 | 0.026 |
> | Ours-P | Transformer Decoder | 0.857($\pm$0.001) | 0.926($\pm$0.001) | 0.770($\pm$0.001) | 0.025($\pm$0.001) |
> | | Convolution Decoder | 0.861($\pm$0.001) | 0.927($\pm$0.001) | 0.776($\pm$0.001) | 0.024($\pm$0.000) |
>
> Note: *std* is rounded to three decimal points. Values below 0.0005 are rounded to 0.000.
>
> Under both convolution-based and transformer-based decoder choices, our method outperforms UAT, attributing the performance gain of Ours-P (with the default convolution-based decoder) equally to the CAReFuse module rather than to the convolution decoder alone.

---

### Review · Reviewer_it6r · 2026-05-08

**Summary Of Contributions:**

This paper introduces CAReFuseNet, a novel and parameter-efficient framework designed for the Referring Camouflaged Object Detection task that identifies camouflaged targets guided by salient reference images. The architecture features a specialized CAReFuse module that leverages cross-attention to align reference features with multi-scale camouflaged image representations within a unified low-dimensional feature space. Extensive experiments on the R2C7K dataset demonstrate that the proposed method achieves state-of-the-art performance while requiring significantly fewer trainable parameters and computational resources than existing heavy transformer-based models.

**Audience:**

Yes

**Audience Explanation:**

This research addresses the emerging task of Referring Camouflaged Object Detection, a specialized and increasingly popular extension of Camouflaged Object Detection that is currently attracting widespread interest within the computer vision community.

**Broader Impact Concerns:**

No concerns on the ethical implications of the work.

**Claims And Evidence:**

Yes

**Claims Explanation:**

The claims regarding state-of-the-art performance and parameter efficiency are convincingly supported by extensive quantitative experiments on the R2C7K dataset, thorough ablation studies of the proposed modules, and qualitative visualizations across various backbone architectures.

**Requested Changes:**

1. Evaluation is primarily restricted to the R2C7K dataset which limits the assessment of the model's robustness and generalization capabilities across more diverse camouflaged environments.

2. The methodology lacks a detailed analysis of the actual inference latency on resource-constrained hardware to fully substantiate the claims of deployment efficiency.

3. While the cross-attention mechanism enables global interactions, the paper does not thoroughly discuss the potential computational overhead when scaling to higher input resolutions.

4. The structural novelty of the Cross-Scale Fusion module is relatively incremental as it builds upon standard feature pyramid and skip-connection concepts used in many segmentation networks.

5. The paper provides limited discussion on specific failure cases such as scenarios involving heavy occlusion or indistinguishable boundaries that still pose challenges for the network.

6. The ablation studies focus heavily on the channel dimension but omit an exploration of the impact of varying the number of attention heads or the depth of the cross-attention blocks.

---

> ### Author Response · Authors · 2026-05-29
> **Rebuttal by Authors**
>
> We thank the reviewer for the comments.
>
> **Requested change 1:** Evaluation is primarily restricted to the R2C7K dataset.
> **Response:** While the other popular datasets for COD research are COD10K, CAMO and NC4K, R2C7K is the only dataset available (containing both camouflaged images and salient ref images) for Ref-COD task. Since, in this paper, we have trained and evaluated our method only for the Ref-COD task but not for a generalised COD setting, we could report the performance on the only available benchmark dataset, i.e., R2C7K. Furthermore, the R2C7K dataset is constructed from the COD datasets COD10K and NC4K.
>
> **Requested change 2:** The methodology lacks a detailed analysis of the actual inference latency.
> **Response:** As recommended by the reviewer, we added the inference speed (in FPS) and memory (in MB) comparison between Ours-P, UAT and R2CNet in Table 9 of Appendix B (page 17).
>
> **Requested change 3:** While the cross-attention mechanism enables global interactions, the paper does not thoroughly discuss the potential computational overhead when scaling to higher input resolutions.
> **Response:** Following the reviewer's recommendation, we analyse how the computational overhead increases when scaling to higher input image resolutions. An increase in the resolution of the input image (both $I^{ref}$ and $I^{camo}$) would result in an increase in the sequence length for $\\phi_E$ and $\\phi_{F_j}$. Specifically, if both sequences expand by a factor of $k$, the overall computational cost increases quadratically by a factor of $k^2$. We added a section discussing in detail how the computational cost scales as the input resolutions increase, in Appendix C (page 18).
>
> **Requested change 4:** The structural novelty of the Cross-Scale Fusion module is relatively incremental as it builds upon standard feature pyramid and skip-connection concepts used in many segmentation networks.
> **Response:** While the Cross-Scale Fusion module leverages the feature pyramid concepts, our contribution lies in demonstrating that a single-stage reference feature fusion in a low-dimensional feature space, followed by multi-scale fusion, paired with a light-weight segmentation decoder, can yield superior performance to existing Ref-COD methods at a fraction of the parameter budget.
>
> **Requested change 5:** The paper provides limited discussion on specific failure cases.
> **Response:** Following the reviewer's suggestions, we included a discussion on the failure cases along with visual examples (Figure 7 on page 19) in Appendix E (page 18).
>
> **Requested change 6:** The ablation studies focus heavily on the channel dimension but omit an exploration of the impact of varying the number of attention heads or the depth of the cross-attention blocks.
> **Response:** Following the reviewer's suggestion, we conducted the ablation study on the number of attention heads used in cross-attention reference fusion, and the results are reported in Table 7 of Appendix A2 (page 17). While Ours-P uses 4 attention heads by default, the depth of cross-attention is restricted to 1 (single-stage fusion) for parameter efficiency.

---

### Review · Reviewer_93Jo · 2026-05-12

**Summary Of Contributions:**

**Summary**

The paper studies referring camouflaged object detection, where the goal is to segment camouflaged objects of a target category using one or more reference images that show salient examples of that category. The authors propose CAReFuseNet, a dual-branch framework that extracts features from reference images and camouflaged images, then fuses them through a cross-attention based module named CAReFuse. The module includes cross-attention fusion between reference features and multi-scale camouflaged image features, followed by multi-scale fusion and a lightweight convolutional decoder. The main design goal is to reduce trainable and inference parameters while keeping strong segmentation accuracy.

**Strength**

The main strength of the paper is that it targets a useful and relatively recent task, and the proposed model appears effective on R2C7K. In Table 1, the PVT-v2 version of CAReFuseNet improves over UAT on all reported metrics while using much fewer trainable parameters, fewer total parameters, and fewer MACs. The paper also evaluates several backbones and includes ablations on the feature dimension, the number of reference images, the reference feature combiner, CAReFuse components, and supervision strategy.

**Weakness**

The main weakness is that the evidence is still narrow. The experiments are mainly on one benchmark, and the strongest performance gain over UAT is modest in some metrics, so variance across random seeds or splits should be reported. The efficiency claim is supported by parameters and MACs, but not by practical inference speed, memory use, or latency under different numbers of reference images. The novelty is also somewhat limited because cross-attention fusion and cross-scale fusion are known tools; the paper’s value is mostly in adapting and simplifying them for Ref-COD rather than introducing a clearly new learning principle.

**Audience:**

Yes

**Audience Explanation:**

The paper should be of interest to part of the TMLR audience, especially readers working on computer vision, segmentation, attention-based architectures, and efficient learning systems. Ref-COD is a meaningful task because it differs from standard camouflaged object detection by conditioning segmentation on a user-specified target category through reference images. The paper also addresses a practical issue: existing methods may be large, while CAReFuseNet gives a simpler and smaller alternative with strong results.

The contribution is not broad across all of machine learning, and the method is mainly an architecture paper for a specific vision task. Still, TMLR’s criteria allow acceptance for work with modest contribution if the claims are supported and at least some of the audience would find the result useful. The paper meets this bar in my view, although the final version should better qualify the scope of its claims and strengthen the empirical analysis.

**Broader Impact Concerns:**

The paper mentions applications of camouflaged object detection in medical image segmentation and video surveillance. The surveillance use case raises possible concerns because better camouflaged person or object detection could be used in monitoring, tracking, or military settings. I recommend adding a short Broader Impact Statement that discusses possible misuse, dataset privacy if any human imagery is used, limits of reliability in safety-critical medical or surveillance settings, and the need for human review before use in high-stakes systems.

**Claims And Evidence:**

Yes

**Claims Explanation:**

The main claims are mostly supported. The paper claims that CAReFuseNet is a parameter-efficient Ref-COD model and that it outperforms prior Ref-COD methods. The quantitative evidence in Table 1 supports this claim. The ablation studies also give reasonable support for the proposed design.

However, the evidence would be more convincing with uncertainty estimates, multiple training runs, and practical speed/memory results. Since the absolute gain over UAT is small for some metrics, a single-run comparison is not enough to fully judge robustness. The paper should also be more careful with the phrase “state-of-the-art” because some comparisons are made under different image sizes and different numbers of reference images. I still answer “Yes” because the main empirical claims are backed by clear tables and ablations, but several parts need stronger reporting.

**Requested Changes:**

1. Add results over multiple random seeds or repeated runs, at least for the main Ours-P model and the strongest baseline comparison with UAT. The gains over UAT are positive but not very large in some metrics, so the paper should report mean and standard deviation, confidence intervals, or another measure of variability. (Critical)

2. Strengthen the efficiency evaluation. Parameter counts and MACs are useful, but they do not fully support deployment-related claims. The paper should report inference latency, peak GPU memory, and throughput under the same hardware and batch size for Ours-P, UAT, and R2CNet. The authors should also clarify whether reference features are computed online or can be cached, and report the cost for different values of K. (Critical)

3. Make the comparison protocol clearer and more fair. The paper separates the CIRCOD comparison because CIRCOD uses different settings, which is good, but the text should avoid unqualified “state-of-the-art” claims unless all compared methods are evaluated under the same input size, number of references, training data, and test split. If possible, rerun UAT, R2CNet, and CIRCOD under a unified setting, or clearly state which numbers are copied from prior papers and which are reproduced by the authors. (Critical)

4. Add a deeper failure analysis. The qualitative results show successful cases on small objects, thin structures, occlusion, and complex boundaries, but the paper should also show representative failure cases. This would help readers understand when the reference image does not give enough guidance, when the model confuses similar categories, and when camouflage is too strong. (not necessarily critical)


5. Evaluate the “target absent” case or clearly mark it as outside the paper’s scope. The limitations section states that the method assumes the camouflaged image contains the target object specified by the references. This assumption is restrictive. Even a small synthetic or held-out evaluation with mismatched references would be useful for measuring false positives and understanding model behavior in more realistic use. (not necessarily critical)


6. Discuss novelty more carefully. The use of cross-attention and multi-scale fusion is not new by itself. The paper should state more precisely what is new: for example, the low-dimensional reference-conditioned fusion design, the single-stage fusion for Ref-COD, the parameter-efficient decoder choice, or the combination of these choices for this task. (not necessarily critical)

---

> ### Author Response · Authors · 2026-05-29
> **Rebuttal by Authors**
>
> We thank the reviewer for the comments.
>
> **Requested change 1:** Add results over multiple random seeds.
> **Response:** Following the reviewer's recommendation, we re-run the experiments of Tables 1, 4, 5, and 6 with 3 different random seeds and reported $mean \pm std$. The tables are updated accordingly. Note: The standard deviation values below $0.0005$ are rounded to $0.000$.
>
> **Requested change 2:** Strengthen the efficiency claim.
> **Response:** Following the reviewer's suggestion, we added the inference speed (in FPS) and memory (in MB) comparison between Ours-P, UAT and R2CNet in Table 9 of Appendix B (page 17). In addition, the inference speed and memory of Our-P for different values of $K$ is reported in Table 10 of Appendix B (page 17). And we clarify that the reference features are computed once and cached rather than computing online.
>
> **Requested change 3:** Make the comparison protocol clearer and more fair.
> **Response:** In addition to using a different image size, CIRCOD, by construction, does not accept multiple reference images; therefore, the unified setting (with 5 referring images) followed by UAT and R2CNet could not be applied to it. And we clarify that all the results of R2CNet and UAT(in Table 1) and the results of CIRCOD (in Table 2) are taken from the respective prior works, while the results of Ours are reported by us. We have now included this information in the captions of Tables 1 and 2  for better clarity.
>
> **Requested change 4:** Add a deeper failure analysis.
> **Response:** Following the reviewer's recommendation, we included a discussion on the failure cases along with visual examples (Figure 7 on page 19) in Appendix E (page 18).
>
> **Requested change 5:** Evaluate the “target absent” case or clearly mark it as outside the paper’s scope.
> **Response:** As mentioned under 'Limitations and Future Directions', the 'target absent' case is indeed outside the scope of the Ref-COD task addressed in our work. Nevertheless, we added a discussion on our model’s predictions for camouflaged images with negative (or mismatched) reference images in Appendix F (page 18). The examples (Figure 8 on page 19) reveal that even with negative reference images, our model still incorrectly produces some false-positive regions in the segmentation masks instead of producing a blank output. The other Ref-COD methods (UAV and R2CNet) also produced false positive regions when presented with negative references.
>
> **Requested change 6:** Discuss the novelty more carefully.
> **Response:** While our method leverages cross-attentions for feature fusion, our contribution lies in demonstrating that a single-stage reference feature fusion in a low-dimensional feature space, paired with a light-weight segmentation decoder, can yield superior performance to existing Ref-COD methods at a fraction of the parameter budget. Following the reviewer's suggestion, we modified the first bullet point of the summary of our contributions (at the end of the Introduction section) in the manuscript.

---

### Review · Reviewer_ByB3 · 2026-05-12

**Summary Of Contributions:**

The paper introduces CAReFuseNet, a highly parameter-efficient architecture for Referring Camouflaged Object Detection. The core technical contribution is the CAReFuse module, which executes cross-attention and multi-scale feature fusion within a constrained, low-dimensional space, followed by a lightweight convolutional decoder. By avoiding heavy transformer-based decoding, the proposed model achieves state-of-the-art performance on the R2C7K dataset while utilizing only a fraction of the parameters required by leading baselines like UAT.

**Strength**

The motivation for parameter efficiency in resource-constrained environments is clear and well-justified.

The experimental section is robust. The authors demonstrate the architecture's generality across multiple well-known backbones, including ResNet, EfficientNet, Swin, ConvNeXt, and PVT.

**Weakness**

The model operates under the rigid assumption that the target object is always present in the camouflaged input image, which limits real-world, unconstrained applicability.

While computational complexity is measured in parameters and Multiply-Accumulate operations, the paper lacks empirical inference speed metrics, which are standard for evaluating efficient deployment.

**Audience:**

Yes

**Audience Explanation:**

The TMLR audience has a strong, ongoing interest in efficient deep learning architectures and practical, low-footprint applications of cross-attention mechanisms. Camouflaged Object Detection and its referring variant have growing real-world utility in impactful domains, ranging from medical image segmentation to wildlife monitoring. Designing visual recognition models that retain competitive performance while significantly lowering the computational footprint is a highly relevant endeavor, directly addressing the deployment constraints of latency-sensitive edge applications.

**Claims And Evidence:**

Yes

**Claims Explanation:**

The authors successfully back their core claims of parameter efficiency and superior accuracy through extensive benchmarking against established Ref-COD baselines, such as R2CNet, UAT, and CIRCOD. Table 1 provides clear, empirical evidence that the proposed CAReFuseNet achieves higher structure-measure, adaptive E-measure, and weighted F-measure, alongside a lower mean absolute error compared to UAT. It does so while significantly reducing both trainable and total parameter counts. Furthermore, the paper includes detailed ablation studies that validate the architectural choices, proving that both the CAF and MSF modules yield measurable performance gains. The qualitative visual results also convincingly substantiate the model's ability to segment challenging targets, such as highly occluded objects and complex object boundaries.

**Requested Changes:**

1. The paper excels at comparing parameter counts and MACs to substantiate its efficiency claims. However, to truly claim the model aids efficient deployment, hardware-level inference speed must be added to the quantitative evaluation section.

2. The qualitative results currently only showcase scenarios where the model outperforms baselines. Please include a dedicated discussion and visual examples of failure modes. Showing instances where the model hallucinates false positives, fails to align with the reference image, or struggles with specific textures will provide readers with a necessary, balanced view of the architecture's limitations.

3. The authors transparently note in the limitations that the model assumes the target is always present in the camouflaged image. I recommend adding a brief pilot experiment or extended discussion detailing how the network behaves when presented with negative samples. This would better reflect practical deployment scenarios.

4. As noted in the conclusion, the model currently processes a single category at a time. Expanding the Future Directions section to discuss potential architectural pathways for multi-category Ref-COD would enrich the paper's theoretical value.

---

> ### Author Response · Authors · 2026-05-29
> **Rebuttal by Authors**
>
> We thank the reviewer for the comments.
>
> **Requested change 1:** The paper excels at comparing parameter counts and MACs to substantiate its efficiency claims. However, to truly claim the model aids efficient deployment, hardware-level inference speed must be added to the quantitative evaluation.
> **Response:** Following the reviewer's suggestion, we added the inference speed (in FPS) and memory (in MB) comparison between Ours-P, UAT and R2CNet in Table 9 of Appendix B (page 17).
>
> **Requested change 2:** The qualitative results currently only showcase scenarios where the model outperforms baselines. Please include a dedicated discussion and visual examples of failure modes.
> **Response:** Following the reviewer's recommendation, we included a discussion on the failure cases along with visual examples (Figure 7 on page 19) in Appendix E (page 18).
>
> **Requested change 3:** The authors transparently note in the limitations that the model assumes the target is always present in the camouflaged image. I recommend adding a brief pilot experiment or extended discussion detailing how the network behaves when presented with negative samples.
> **Response:** Following the reviewer's recommendation, we added the discussion on our model's predictions for camouflaged images with negative (or mismatched) reference images in Appendix F (page 18). The examples (Figure 8 on page 19) reveal that even with negative reference images, our model still incorrectly produces some false-positive regions in the segmentation masks instead of producing a blank output. The other Ref-COD methods (UAV and R2CNet) also produced false positive regions when presented with negative references.
>
> **Requested change 4:** As noted in the conclusion, the model currently processes a single category at a time. Expanding the Future Directions section to discuss potential architectural pathways for multi-category Ref-COD would enrich the paper's theoretical value.
> **Response:** As recommended by the reviewer, we extended our Future Directions section discussing a potential pathway to support multi-category reference images for the Ref-COD task.
> A potential architectural pathway to achieve this involves a two-stage adaptation: first, the model must be adapted to output a null (blank) segmentation mask when presented with mismatched or negative reference images. Once this capability is achieved, multi-category support can be realised by instantiating parallel streams of our CAReFuse module - one for each reference category - to individually align the camouflaged image features with their respective reference features. These category-specific aligned features can then be processed through a shared segmentation decoder to yield distinct, parallel prediction masks. Finally, these individual predictions can be combined into a comprehensive multi-category segmentation mask.